# A Three-Year Longitudinal Study Comparing Bone Mass, Density, and Geometry Measured by DXA, pQCT, and Bone Turnover Markers in Children with PKU Taking L-Amino Acid or Glycomacropeptide Protein Substitutes

**DOI:** 10.3390/nu13062075

**Published:** 2021-06-17

**Authors:** Anne Daly, Wolfgang Högler, Nicola Crabtree, Nick Shaw, Sharon Evans, Alex Pinto, Richard Jackson, Catherine Ashmore, Júlio C. Rocha, Boyd J. Strauss, Gisela Wilcox, William D. Fraser, Jonathan C. Y. Tang, Anita MacDonald

**Affiliations:** 1Birmingham Women’s and Children’s Hospital, NHS Foundation Trust, Birmingham B4 6NH, UK; nicola.crabtree@nhs.net (N.C.); nick.shaw@nhs.net (N.S.); evanss.21@me.com (S.E.); alex.pinto@nhs.net (A.P.); catherine.ashmore@nhs.net (C.A.); anita.macdonald@nhs.net (A.M.); 2Department of Paediatrics and Adolescent Medicine, Johannes Kepler University, Kepler University Hospital, Krankenhausstraße 26-30, 4020 Linz, Austria; wolfgang.hoegler@kepleruniklinikum.at; 3Liverpool Clinical Trials Centre, University of Liverpool, Brownlow Hill, Liverpool L69 3GL, UK; r.j.jackson@liverpool.ac.uk; 4Nutrition and Metabolism, NOVA Medical School, Faculdade de Ciências Médicas, Universidade Nova de Lisboa, 1169-056 Lisboa, Portugal; rochajc@nms.unl.pt; 5Centre for Health and Technology and Services Research (CINTESIS), 4200-450 Porto, Portugal; 6School of Medical Sciences, Faculty of Biology, Medicine and Health Sciences, University of Manchester, Manchester M13 9PL, UK; boyd.strauss@manchester.ac.uk (B.J.S.); gisela.wilcox@manchester.ac.uk (G.W.); 7School of Clinical Sciences, Faculty of Medicine, Nursing and Health Sciences, Monash University, Melbourne 3800, Australia; 8The Mark Holland Metabolic Unit, Salford Royal Foundation NHS Trust, Ladywell NW2, Salford, Manchester M6 8HD, UK; 9BioAnalytical Facility, BCRE Builiding University or East Anglia, Norwich NR4 7TJ, UK; Fraser@uea.ac.uk (W.D.F.); Jonathan.Tang@uea.ac.uk (J.C.Y.T.); 10Departments of Clinical Biochemistry and Endocrinology, Norfolk and Norwich University Hospitals Trust, Norwich NR4 7UY, UK

**Keywords:** PKU, bone mineral density, bone turnover markers, osteoporosis, blood biochemistry, casein glycomacropeptide, amino acid protein substitute

## Abstract

In patients with phenylketonuria (PKU), treated by diet therapy only, evidence suggests that areal bone mineral density (BMDa) is within the normal clinical reference range but is below the population norm. Aims: To study longitudinal bone density, mass, and geometry over 36 months in children with PKU taking either amino acid (L-AA) or casein glycomacropeptide substitutes (CGMP-AA) as their main protein source. Methodology: A total of 48 subjects completed the study, 19 subjects in the L-AA group (median age 11.1, range 5–16 years) and 29 subjects in the CGMP-AA group (median age 8.3, range 5–16 years). The CGMP-AA was further divided into two groups, CGMP100 (median age 9.2, range 5–16 years) (*n* = 13), children taking CGMP-AA only and CGMP50 (median age 7.3, range 5–15 years) (*n* = 16), children taking a combination of CGMP-AA and L-AA. Dual X-ray absorptiometry (DXA) was measured at enrolment and 36 months, peripheral quantitative computer tomography (pQCT) at 36 months only, and serum blood and urine bone turnover markers (BTM) and blood bone biochemistry at enrolment, 6, 12, and 36 months. Results: No statistically significant differences were found between the three groups for DXA outcome parameters, i.e., BMDa (L2–L4 BMDa g/cm^2^), bone mineral apparent density (L2–L4 BMAD g/cm^3^) and total body less head BMDa (TBLH g/cm^2^). All blood biochemistry markers were within the reference ranges, and BTM showed active bone turnover with a trend for BTM to decrease with increasing age. Conclusions: Bone density was clinically normal, although the median z scores were below the population mean. BTM showed active bone turnover and blood biochemistry was within the reference ranges. There appeared to be no advantage to bone density, mass, or geometry from taking a macropeptide-based protein substitute as compared with L-AAs.

## 1. Introduction

Optimal bone mass is key to preventing the risk of fractures later in life, and many factors influence peak bone mass accretion including genetics, physical activity, body composition, and quality of diet. Severe dietary restriction may be problematic in conditions such as phenylketonuria (PKU) which require rigorous exclusion of many natural foods [1]. In children with classical PKU, the majority of protein is provided by a low phenylalanine, semisynthetic protein (protein substitute), with some limited dietary phenylalanine given from natural foods according to individual metabolic tolerance and disorder severity. Dependency on a synthetic protein may compromise both peak bone mass attainment and bone geometry [2,3].

Protein substitutes, are traditionally derived from essential and non-essential amino acids and are usually supplemented with added vitamins, minerals, and trace minerals aimed at achieving optimal growth, bone mass, and body composition. Protein substitutes are necessary lifelong, but long-term adherence is difficult to sustain particularly during adolescence [4,5], which is a vulnerable time for maximising bone mass, density, mineralization, and growth potential. Amino acids (AAs) contribute to the structural components of bone in addition to those of growth and tissue maintenance [2,6,7].

Protein has a positive effect on bone [6,7], and protein intake promotes peripubertal bone growth and delays bone loss [8,9]. Several long-term prospective observational studies [10,11] have shown significant positive associations between protein intake and bone mineral content, periosteal circumference, cortical area, and an index of strength strain. These studies reinforce that a moderate to high protein diet promotes bone accretion. The acid ash theory suggests that a high protein intake including protein substitutes based on amino acids are detrimental to bone accretion [8,12]. Protein substitutes are acidic, producing sulphuric acid from sulphur containing amino acids. The hypothesis suggests that calcium stored primarily in bones is slowly excreted to buffer the acidic pH, and this process leads to a decreased bone mineral density [13,14,15,16]. However, systematic and meta-analysis studies have dismissed this theory [17,18]. Although the urine pH is lower when taking a protein rich diet, the pH of the extracellular fluid is undisturbed due to regulatory control by the kidneys [8].

The use of casein glycomacropeptide supplemented with amino acids (CGMP-AA) has been associated with improved bone mass in PKU animal models [19], but CGMP (a bioactive peptide) compared with AAs and their influence on bone mass, density, and geometry has not been studied in children with PKU.

In this longitudinal prospective controlled study over 36 months, we investigated the efficacy of CGMP-AA as compared with L-AA protein substitutes on bone mass, density, geometry, and turnover markers in children with PKU.

## 2. Materials and Methods

### 2.1. Methods

The inclusion criteria included the following: children with PKU diagnosed by newborn screening, children aged 5–16 years and not treated with sapropterin dihydrochloride, known adherence with protein substitutes, and maintenance of 70% of blood phenylalanine concentrations within the European PKU target therapeutic range for 6 months prior to study enrolment [20]. Target blood phenylalanine ranges for children aged 5–12 years were from 120 to ≤360 µmol/L, and for children 12 years and older they were from 120 to ≤600 µmol/L.

#### 2.1.1. Ethical Approval

This study was registered by the Health Research Authority and was given a favourable ethical opinion by the South Birmingham Research Ethical Committee (referenced 13/WM/0435 and IRAS (integrated research application system) number 129497). Written informed consent was given by at least one caregiver with parental responsibility and written consent was obtained from the subjects if appropriate for their age and level of understanding.

#### 2.1.2. CGMP-AA and L-AA Protein Substitutes

The CGMP-AA (a test product by Vitaflo International Ltd., Liverpool, UK) was a flavoured powder. Each 35 g sachet contained 20 g protein equivalent, and 36 mg phenylalanine, mixed with 120 mL of water. The flavoured L-AA was either a powder mixed with water or a ready-prepared liquid that provided 10, 15, or 20 g of protein equivalent. The CGMP-AA and L-AA products both had a similar nutritional and AA profile, except CGMP-AA contained residual phenylalanine and higher amounts of threonine and leucine.

#### 2.1.3. Selection into the CGMP Group or L-AA Group

The children chose the product they preferred, depending on their taste preference, i.e., the CGMP-AA group or L-AA group. They remained on this formula for the duration of the study.

### 2.2. Study Design

The primary aim of this 3-year longitudinal study was to compare bone mass, density and geometry of children with PKU taking CGMP-AA or L-AA as their primary protein source. The following examinations were conducted: dual-energy X-ray absorptiometry (DXA), together with blood bone biochemistry and blood and urine bone turnover markers. Peripheral quantitative computed tomography (pQCT) of the forearm was performed at 36 months only (Figure 1 and Table 1).

A previous pilot study [21] demonstrated that the residual phenylalanine in the CGMP-AA group led to compromised phenylalanine control in some children. Therefore, the CGMP-AA group was subdivided into: (1) CGMP100 group, in which the children took the entire protein substitute as CGMP-AA and (2) CGMP50 group, in which children took a combination of L-AA and CGMP-AA. There was also a third group of children who remained on their usual L-AA only (L-AA group).

#### 2.2.1. Dual-Energy X-ray Absorptiometry (DXA) and Peripheral Quantitative Computed Tomography (pQCT)

A GE Lunar iDXA and Encore™ software version 13.1 g (GE Healthcare, Madison, WI, USA) was used to measure bone density at enrolment and at the end of 36 months. Trunk thickness and body weight were used as a guide for scanning each child in the most appropriate acquisition mode. Children lay supine on a bed, while the DXA scan was completed. The following measurements were performed: lumbar spine (L2–L4) areal bone mineral density (L2–L4 BMDa) in g/cm^2^, lumbar spine (L2–L4) bone mineral content (L2–L4 BMC) in g, total body mineral content (BMC) in g, total body less head BMDa (TBLH) in g/cm^2^, and size corrected outcome measures included lumbar spine bone mineral apparent density (L2–L4 BMAD) in g/cm^3^. At 36 months, in addition to the DXA assessment, pQCT was also performed.

#### 2.2.2. pQCT

The pQCT (Stratec XCT 2000 L, Pfozheim, Germany) measurements were taken at the 4% and 66% region of the non-dominant forearm, evaluating volumetric bone mineral density, together with muscle and bone geometry, size, and strength. At the 4% site, trabecular and total cross-sectional area were measured, while at the 66% site, cortical density, as well as muscle, bone, and fat area were measured. The pQCT also measured the strength strain index as a surrogate marker of bone strength.

#### 2.2.3. Serum Blood and Urine Bone Turnover Markers

Fasting, early morning, venous blood samples were collected at enrolment, 6, 12, and 36 months for the following serum bone markers: procollagen type 1 N-terminal propeptide (P1NP), type 1 collagen β crosslinked C-telopeptide (β-CTX), and bone alkaline phosphatase (bone ALP). A urine sample, the second sample of the day, was collected at enrolment, 6, 12, and 36 months for urine creatinine adjusted free urine pyridinoline (fPYD/Ur Cr) and urine free deoxypyridinoline crosslinks (fDPD/Ur Cr), and urinary calcium/creatinine ratio (Ur Ca/Cr). Urine samples were collected in containers, which were wrapped in tin foil and put into an envelope to shield them from any light. All urine samples were taken immediately to the laboratory for processing and stored at −80 °C. β-CTX and P1NP were analysed using an electro-chemiluminescence immunoassay (ECLIA) on a COBAS e601 analyser (Roche Diagnostics, Mannheim, Germany). The inter-assay coefficient of variation (CV) for β-CTX was <3% across the analytical range, between 0.01 and 6.0 µg/L, with a sensitivity of 0.01 µg/L. The inter-assay CV for P1NP was <3%, between 5 and 1200 μg/L, with a sensitivity of 5 µg/L. The serum bone ALP was determined by MicroVue™ enzyme-linked immunosorbent assay ELISA kit (Quidel Corporation, San Diego, USA). The inter-assay CV for bone ALP was <5.8%, between 0.5 and 150 U/L, with a detection limit of 0.7 U/L.

The analyses for urinary fPYD and fDPD were performed using the liquid chromatography tandem mass spectrometry (LC-MS/MS) method, as described by Tang et al [22]. In brief, 0.5 mL of urine sample/calibration/quality control materials pretreated with 0.5 mL hydrochloric acid (40% concentrate) was extracted using a solid phase extraction (SPE) column packed with cellulose slurry. Pyridinium crosslinks were eluted from the SPE columns and analysed by LC–MS/MS coupled with an electrospray ionisation (ESI) source operated in positive mode. The inter-assay CVs were ≤10.3% for PYD in the concentration range of 5–2000 nmol/L and ≤13.1% for DPD between 2 and 1000 nmol/L. The lower limit of quantification was 6 nmol/L for fPYD and 2.5 nmol/L for fDPD.

Urine creatinine was measured to obtain the fPYD/ and fDPD/urine creatinine ratios and the urine calcium/creatine ratio. Samples were analysed using Roche kinetic colorimetric assays performed on a COBAS^®^ C501 analyser (Roche, Burgess Hill, UK), according to the manufacturer’s instructions. The inter-assay CV ranged from 1.3 to 2.1% across the assay working range for Ur Ca of 0.20–7.5 mmol/L and Ur creatinine of 0.355 mmol/L.

#### 2.2.4. Blood Biochemistry Markers

Overnight fasting blood samples for serum calcium, magnesium, phosphate, vitamin D, and parathyroid hormone were collected at enrolment, 6, 12, and 36 months.

#### 2.2.5. Blood Phenylalanine/Tyrosine Monitoring

Throughout the 36-month study, trained caregivers collected weekly overnight fasting morning blood spots at home for phenylalanine and tyrosine. Blood specimens were sent via the post to the Birmingham Women’s and Children’s Hospital Laboratory. The blood spot filter cards used were Perkin Elmer 226 UK standard NBS (Perkin Elmer, Waltham, MA, USA). All the cards had a standard thickness, and the blood phenylalanine and tyrosine concentrations were calculated on a 3.2 mm punch by tandem mass spectrometry.

#### 2.2.6. Pubertal Status

A general medical examination and pubertal status was measured at enrolment using the Tanner picture index. Stages 1 and 2 are classified as pre-pubertal, and Stages 3, 4, and 5 are classified as pubertal.

#### 2.2.7. Anthropometric Measurements

Weight and height were measured once every 3 months by one of two metabolic dietitians. Height was measured using a Harpenden stadiometer (Holtain Ltd., Crymych, Wales, UK).

### 2.3. Statistical Methods

Continuous data are presented as median and interquartile ranges and categorical data are presented as frequencies of counts with associated percentages. Longitudinal data are presented graphically using profile plots to show the average change over time. Correlations between continuous covariates were evaluated using Pearson’s correlation coefficient. Comparisons between treatment groups were performed using analysis of covariance (ANCOVA) techniques, to analyse the follow-up data, while including baseline measures as adjusting covariates. Models also included covariates for patients’ gender, age, and puberty status (supplementary data are provided for these parameters). A *p*-value of 0.05 was used throughout to determine statistical significance. All analyses were performed using R (Version 3).

## 3. Results

### 3.1. Subjects

Fifty children (28 boys and 22 girls) with PKU were recruited. Forty-seven children were of European origin and three children were of Asian origin. Forty-eight children completed the study, 29 children in the CGMP-AA group and 19 children in the L-AA group. At enrolment, the median age (range) in the CGMP100 group was 9.2 years (5–16 years) (*n* = 13); in the CGMP50 group, the median age was 7.3 years (5–15 years) (*n* = 16), and in the L-AA group, the median age was 11.1 years (5–16 years) (*n* = 19). Only six children were able to tolerate >10 g/day of natural protein (CGMP100 *n* = 2, CGMP50 *n* = 1, and L-AA *n* = 3), all the others received <10 g/day of natural protein.

#### 3.1.1. Subject Drop Out

One boy and one girl (both aged 12 years) in the CGMP-AA group were excluded from the study as both failed to comply with the study protocol. One failed to return blood phenylalanine samples and both had poor adherence to the low phenylalanine diet.

#### 3.1.2. Pubertal Status

The number of children prepubertal (Stages 1 and 2) at enrolment were: CGMP100 group, 62% (*n* = 8/13); CGMP50 group, 69% (*n* = 11/16); and L-AA group, 32% (*n* = 6/19).

The number of children in puberty (Stages 3 to 5) were: CGMP100 group, 38% (*n* = 5/13); CGMP50 group, 31% (*n* = 5/16); and L-AA group, 68% (*n* = 13/19).

#### 3.1.3. Median DXA Z Score Measurements for CGMP100, CGMP50, and L-AA Groups 

Overall, there were no significant differences among the groups for any of the measured DXA parameters. Bone density was on the lower side of normal but within a normal reference range (Table 2).

#### 3.1.4. Median pQCT Z Score Measurements at 36 Months for CGMP100, CGMP50, and L-AA Groups

Similar to the DXA z score measurements, overall, there were no significant differences among the groups, but cortical density at the 66% site was statistically significantly different between the CGMP100 and L-AA groups (Table 3).

### 3.2. Nutritional Bone Biochemistry Markers

Median concentrations for all the biochemistry markers (calcium, phosphate, magnesium, vitamin D, and parathyroid hormone) were within normal reference ranges for all the groups over the 36-month study period (Table 4). There were no statistically significant differences within or among the groups.

#### Measurement for Bone Formation Markers and Urine Calcium

The urine calcium/creatinine ratio (Ur Ca/Cr) a measure of renal acid excretion was normal with no indication of excess calcium excretion (Table 5). Similarly, serum and urine BTM showed a physiological decrease with age, and no evidence of a disturbance between formation and resorption.

A strong positive correlation was observed between PN1P and β-CTX at 36 months (*r* = 0.82) (Figure 2). The ANCOVA analysis performed on PN1P indicated that the level of PN1P was somewhat dependent on age, with older subjects having a lower PN1P level. Furthermore, there was evidence of an increase in PN1P at 36 months associated with CGMP100 as compared with L-AA (*p* = 0.041) (Figure 3). There was no difference between the CGMP50 and L-AA groups (*p* = 0.80).

### 3.3. Anthropometry

We have previously reported height, weight, and body mass index in this group of children [23]. At 36 months, all groups had a median positive height z score: L-AA, 0.2 (range 0 to 0.5); for CGMP50, 0.3 (range −0.1 to 0.7); and for CGMP100, 0.6 (range 0.1 to 0.7). Median weight for height z scores and BMI z scores were above the ideal reference mean, indicating an overweight group of children (Table 6).

### 3.4. Blood Phenylalanine Concentrations

The median phenylalanine concentrations for this study have been previously reported. Median phenylalanine concentrations were within recommended target reference ranges for children aged ≤11 and ≥12 years old [23].

The median daily dose of protein equivalent from protein substitute was 60 g/day (range 40–80 g), and the median amount of prescribed natural protein was 5.5 g protein/day (range 3–30 g) or 275 mg/day of phenylalanine (range 150–1500 mg), in all three groups. Eighty-eight percent (*n* = 42) of the children tolerated ≤10 g/day natural protein and 12% (*n* = 6) >10 g/day (CGMP100, *n* = 2; CGMP50, *n* = 1; and L-AA, *n* = 3).

## 4. Discussion

In this 36-month longitudinal study in children with PKU, bone mass, density, and geometry were comprehensively examined by DXA and pQCT, in addition to serum BTM and blood biochemistry. With the exception of cortical density at the 66% site, none of the other bone measurements showed any benefit of CGMP100 over L-AA or CGMP50, suggesting that CGMP-AA had no advantage over L-AA for bone development. Similarly, there was no evidence to suggest any differences in bone mass, density, or geometry by gender, age, or puberty (Appendix A).

A strong positive correlation between β-CTX and P1NP was observed in all three study groups, with P1NP being lower in the older age subjects, and an increased P1NP being evident in the CGMP100 group. This synergy between bone formation and resorption shows active bone turnover and reflects appropriate bone growth, since these markers derive from physiological processes. Our results contrast with those reported by Casto et al. [24], which suggested a trend towards increased bone resorption in subjects with PKU. This controlled study, was the first to monitor bone mass and density using two separate imaging technologies (DXA and pQCT), and holistically assesses serum bone, urine, and blood biochemistry parameters in PKU. Similar to findings from two systematic reviews [24,25], the overall bone density values for the groups were below the population mean but within the normal reference values. Imaging results met the International Society for Clinical Densitometry (ISCD) recommendations (ISCD 2013) [26]. There were no differences in biochemical or BTM among the groups, suggesting no changes in bone metabolism attributed to the type of protein substitute. Naturally, BTM concentrations decreased in older adolescents towards those of lower adult levels, as a physiological phenomenon expected in a healthy population [27].

Unlike the findings of Schwahn et al., Mc Murry et al., and Fernandez et al. [28,29,30], we found no evidence to suggest that mineralization defects began in childhood, and then became more evident in adolescents. In this study, the groups of children were overweight. The relationship between overweight, obesity and bone is contentious.

Evidence [31] suggests that in early childhood obesity confers a structural advantage on the developing skeleton, but with age this relationship is reversed and becomes detrimental to skeletal development. Clarke et al [32] reported a positive relationship between adiposity and bone mass accrual in 3082 healthy children, while others [33,34] have reported opposite findings. Lean body mass has been shown to be the strongest predictor of bone mineral content [35,36] and relates to bone mass and skeletal development in children. Our previous study [37] indicated a trend towards improved lean body mass in the CGMP100 group; however, there was no evidence to suggest a similar beneficial effect on bone density in this group.

In PKU mouse models, CGMP as compared with L-AA has been shown to increase bone strength measured by biochemical mechanisms. Solverson [19] gave PKU and wild type mice different dietary regimens, i.e., a normal diet or a low phenylalanine diet supplemented with L-AA or CGMP protein substitutes. The PKU mice, regardless of protein substitute type, had lower bone density as compared with wild type mice, and those taking L-AA had inferior bone strength as compared with the CGMP protein substitute group. The authors proposed that the peptide structure of CGMP could possibly account for the positive influence on bone radial size improving biochemical performance. Alternatively, the high acid load due to L-AA could decrease bone strength via excreting higher amounts of calcium. However, both these suggestions were conjecture, as they did not measure net acid excretion, bone collagen, and markers of bone biomechanical performance. The results from our study in our cohort of children would suggest that neither of these mechanisms are active. BTM monitoring collagen were physiologically normal and there was no evidence of net acid excretion with a normal calcium/creatinine ratio.

Although many studies have identified lower BMD in PKU [38,39,40,41], not all of these studies included a size correlation for DXA output and there has been little agreement about lower BMD pathophysiology. Dobrowolski et al. [42] studied bone mineralization in PKU mice and showed phenylalanine toxicity inhibited bone mineralization. However, in human studies, there is a discord on the link between hyperphenylalaninemia and bone mass, with some studies showing a correlation and others not [38,40,43,44].

Within the three groups (CGMP100, CGMP50, and L-AA) there were expected physiological changes in the concentrations of BTM. In adults, BTM mainly represent bone remodelling; in children, BTM are released during bone remodelling, modelling, and perpendicular growth. Millet et al. [44] measured urine DPD and bone ALP in patients with PKU and compared these with a healthy paediatric group; bone remodelling was active in children with PKU aged 7–14 years, and bone ALP, as expected, was found to be significantly lower in the oldest group of patients (aged >18 years), although significantly higher DPD concentrations independent of age were reported. In our study, bone resorption and formation markers were consistently lowest in the L-AA group, particularly noticeable in the L-AA girls who had reached late puberty with a median age of 17 (8–18 years) at 36 months [27,43,45,46]. In contrast, the youngest group of CGMP50 boys showed an increase in BTM over the 36 months.

The interpretation of BTM is difficult and their concentrations vary widely in children, affected by a multitude of factors including age, gender, puberty, growth velocity, the rate of mineral accrual, hormonal regulation, nutritional status, circadian, and even day-to-day changes [47]. Paediatric reference data are available for some BTM [48,49,50,51], although UK specific data are lacking, which hampers appropriate interpretation. Specificity for bone tissue as well as sensitivity and specificity of the measurement assays lead to variations, rendering comparisons among study groups difficult [50,52]. Despite these challenges, in our study in which children were followed for 36 months, BTM followed the expected variations for age with no differences between the groups. These children had an active bone turnover profile, supportive of a normal bone mineral density. The reason why their bone mineral densities were below the population median was unclear, but these groups were not at any increased clinical risk of fractures.

There are limitations to this study. Patient numbers in each group were small which reduced the power of this study. An extended follow-up period of >3 years may be needed for any differences to emerge between protein substitute sources, as noted, P1NP was increased in the CGMP100 group. We also did not have a healthy control group, which would have been beneficial to compare differences with the children with PKU. The ages of the children were significantly different in all three groups, and CGMP was given at two different concentrations making any absolute differences difficult to recognize, although statistical modelling was used to account for this variable. Age influences bone changes and children entered puberty over the study period. In children, no bone marker is specific for any of the three different biological processes of modelling, remodelling, and changes in endochondral ossification. However, our findings were consistent, i.e., all measurements were taken via DXA or pQCT and showed a below average bone density, with no significant differences among the groups taking CGMP-AA or L-AA. Bone markers appeared to follow a similar pattern to that in healthy children. We did not measure exercise activity in these groups of children, but a high proportion (60%) participated in regular activities such as football, dancing, and gymnastics.

## 5. Conclusions

In this detailed and comprehensive study measuring global bone development, using both two- and three-dimensional imaging in addition to serum BTM and blood biochemistry, a complete assessment of bone mass, density, geometry, and bone turnover was conducted. There were no statistical differences in the groups of children, who had good metabolic control when taking either L-AA or CGMP-AA protein substitutes. Bone density was normal and similar to the findings from systematic reviews, which suggests it was lower than the population norm but carried no increased osteoporotic risk. Bone remodelling processes appear to be active in children with PKU, with both L-AA and CGMP-AA protein substitutes supporting normal bone growth.

## Figures and Tables

**Figure 1 nutrients-13-02075-f001:**
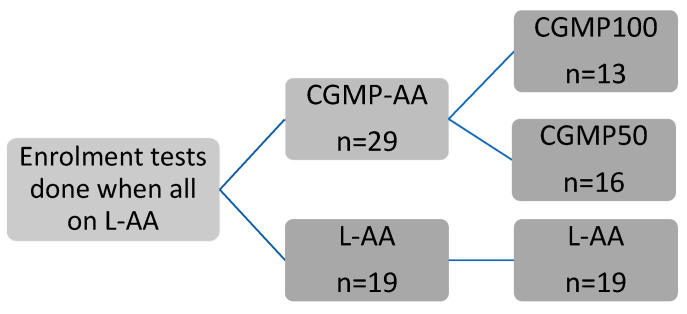
Diagram of the scheme for study methodology, from enrolment to 36 months. Legend: CGMP, casein glycomacropeptide; CGMP100, children taking all their protein substitute as casein glycomacropeptide; CGMP50, children taking a combination of casein glycomacropeptide and amino acids; L-AA, amino acids.

**Figure 2 nutrients-13-02075-f002:**
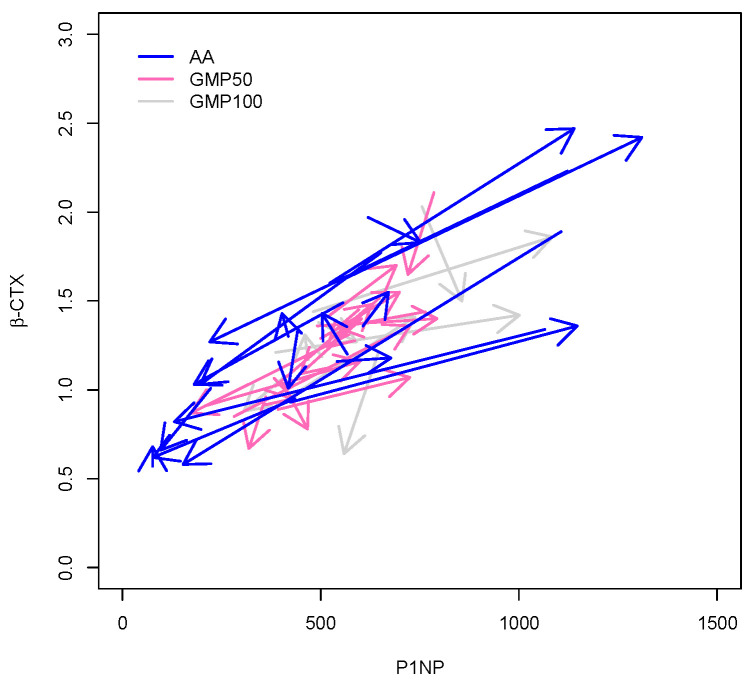
Correlation of β-CTX with PINP for CGMP100, CGMP50, and L-AA, at 36 months (

 CGMP100, glycomacropeptide only; 

 CGMP50, combination of CGMP and L-AA; and 

 L-AA only).

**Figure 3 nutrients-13-02075-f003:**
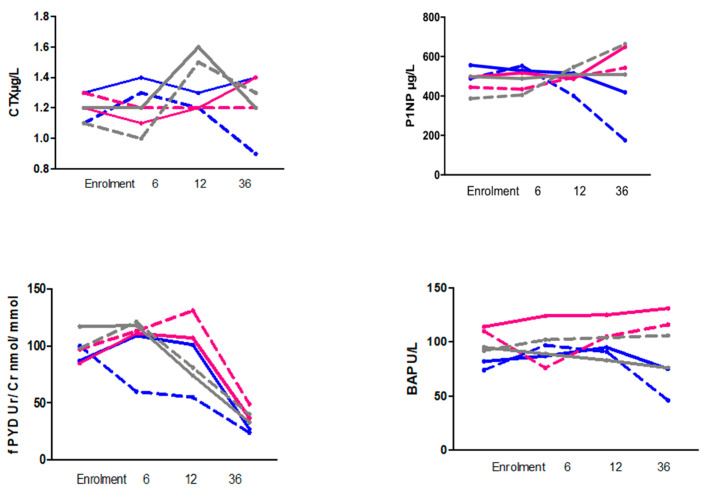
Graphs showing serum and urine bone turnover markers at enrolment, 6, 12, and 36 months separated by gender for CGMP100, CGMP50, and L-AA groups.

**Table 1 nutrients-13-02075-t001:** Frequency of nutritional blood biochemistry, bone blood and urine markers, DXA and pQCT scans, over study duration from enrolment to 36 months.

Enrolment	6 Months	12 Months	36 Months
Fasting blood biochemistry Serum bone markers 2nd void urine bone markers DXA	Fasting blood biochemistry Serum bone markers 2nd void urine bone markers	Fasting blood biochemistry Serum bone markers 2nd void urine bone markers	Fasting blood biochemistry Serum bone markers 2nd void urine bone markers DXA pQCT
**Anthropometry: 3/month**			
**Blood phenylalanine: weekly**			

Legend: CGMP, casein glycomacropeptide; CGMP100, children taking all their protein substitute as casein glycomacropeptide; CGMP50, children taking a combination of casein glycomacropeptide and amino acids; L-AA, amino acids; DXA, dual-energy x-ray absorptiometry; pQCT, peripheral quantitative computerised tomography.

**Table 2 nutrients-13-02075-t002:** Median z scores (range) for L2–L4 bone mineral density (BMDa), lumbar spine bone mineral apparent density (L2–L4 BMAD), and total body less head BMDa (TBLH). Other parameters measured include median (range) L2–L4 bone mineral content and total bone mineral content for CGMP100, CGMP50, and L-AA groups, at enrolment and 36 months.

Group	Enrolment z Score (Range)	36 Months z Score (Range)
**L2–L4 BMDa (g/ cm^2^)**		
CGMP100	−0.2 (−0.9 to 0.8)	−0.6 (−0.9 to 0.6)
CGMP50	−0.1 (−0.5 to 0.5)	−0.1 (−0.6 to 0.4)
L-AA	−0.1 (−0.7 to 0.4)	−0.5 (−0.8 to 0.0)
**L2–L4 BMAD (g/cm^3^)**		
CGMP100	0.2 (−0.9 to 0.6)	0.2 (−0.4 to 0.5)
CGMP50	−0.2 (−0.5 to 0.9)	−0.2 (−0.4 to 0.3)
L-AA	−0.3 (−0.8 to 0.4)	−0.6 (−1.2 to −0.1)
**TBLH BMDa (g/cm^2^)**		
CGMP100	−0.6 (−1 to −0.5)	−0.5 (−0.6 to −0.2)
CGMP50	−0.8 (−1.3 to −0.1)	−0.6 (−0.9 to −0.3)
L-AA	−0.2 (−0.5 to 0.1)	−0.2 (−0.4 to −0.1)
**Median values (range) for Total and L2–L4 BMC g**		
**Total body BMC g**		
CGMP100	832.8 (672.9 to 1543.5)	1258.4 (1082.8 to 1816.9)
CGMP50	604.9 (532.9 to 680.3)	1019.1 (963.4 to 1134.8)
L-AA	1183.8 (672.9 to 1543.5)	1650.2 (1082.8 to 1816.9)
L2–L4 BMC g		
CGMP100	18.9 (14.1 to 22.9)	28.1 (24.1 to 38.3)
CGMP50	14.2 (13.0 to 16.6)	22.1 (20.4 to 25.1)
L-AA	25.6 (15.9 to 34.9)	40.2 (25.0 to 45.4)

Legend: CGMP, casein glycomacropeptide; CGMP100, children taking all their protein substitute as casein glycomacropeptide; CGMP50, children taking a combination of casein glycomacropeptide and amino acids; L-AA, amino acids; L2–L4 BMD, bone mineral density lumbar vertebrae 2 to 4; BMAD, bone mineral apparent density; TBLH, total body less head; L2–L4 BMC, bone mineral content lumbar vertebrae 2 to 4; TBMC, total bone mineral content.

**Table 3 nutrients-13-02075-t003:** Results from the pQCT scan measuring median z scores (range) for trabecular, cortical, and total densities at the 4% site; bone, muscle, and fat areas; strength strain index; and bone area/muscle area at 36 months in the CGMP100, CGMP50, and L-AA groups.

Group	36 Months Z Score (Range)
**Trabecular density: 4%**	
CGMP100	−1.0 (−1.3 to −0.5)
CGMP50	−1.0 (−1.2 to −0.7)
L-AA	−0.5 (−1.2 to −0.1)
**Total density: 4%**	
CGMP100	−0.7 (−1.1 to −0.6)
CGMP50	−0.7 (−0.9 to −0.3)
L-AA	−0.4 (−0.9 to 0.5)
**Cortical density: 66%**	
CGMP100	0.1 (−0.1 to 0.3) *
CGMP50	−0.5 (−1.4 to −0.1)
L-AA	−0.4 (−1.0 to 0.5)
**Bone area: 66%**	
CGMP100	1.9 (1.4 to 4.0)
CGMP50	0.9 (0.2 to 1.8)
L-AA	2.0 (1.5 to 3.7)
**Muscle area: 66%**	
CGMP100	−1.1 (−1.8 to −0.5)
CGMP50	−1.2 (−1.4 to −0.6)
L-AA	−1.0 (−1.8 to −0.5)
**Fat area: 66%**	
CGMP100	0.5 (−0.3 to 0.9)
CGMP50	1.0 (0.4 to 1.8)
L-AA	1.2 (0.1 to 2.3)
**Bone area/muscle area: 66% area**	
CGMP100	0.5 (0.2 to 1.1)
CGMP50	−0.4 (−1.2 to 0.5)
L-AA	0.5 (0.2 to 1.6)
**Strength strain index (SSI): 66%**	
CGMP100	−0.7 (−1.0 to 1.3)
CGMP50	−0.1 (−0.6 to 0.5)
L-AA	0.4 (−0.3 to 0.6)

* CGMP100 as compared with L-AA (*p* = 0.05). Legend: CGMP, casein glycomacropeptide; CGMP100, children taking all their protein substitute as casein glycomacropeptide; CGMP50, children taking a combination of casein glycomacropeptide and amino acids; L-AA, amino acids.

**Table 4 nutrients-13-02075-t004:** Median (range) biochemical bone markers at enrolment and 36 months for CGMP100, CGMP50, and L-AA groups.

	Calcium mmol/L		Phosphate mmol/L		Magnesium mmol/L		25 (OH) Vit D nmol/L		PTH ng/L	
	(Range)		(Range)		(Range)		(Range)		(Range)	
	Enrolment	36 m	Enrolment	36 m	Enrolment	36 m	Enrolment	36 m	Enrolment	36 m
**CGMP100**	2.5	2.4	1.4	1.3	0.9	0.8	112	79	17	32
	(2.3, 2.6)	(2.3, 2.5)	(1.0, 1.5)	(1.0, 1.5)	(0.7, 1.0)	(0.8, 0.9)	(81, 162)	(43.7, 113)	(11, 42)	(22, 57)
**CGGMP50**	2.5	2.4	1.4	1.3	0.8	0.8	94.6	95.2	15.5	31
	(2.3, 2.6)	(2.3, 2.5)	(1.1, 1.6)	(1.1, 1.5)	(0.8, 1.0)	(0.8, 0.9)	(61.8, 135)	(56.3, 137)	(6, 37)	(19, 46)
**L-AA**	2.5	2.4	1.3	1.2	0.8	0.8	93.9	91.8	21	31
	(2.3, 2.6)	(2.3, 2.5)	(1.0, 1.5)	(0.8, 1.7)	(0.8, 0.9)	(0.7, 0.9)	(38.8, 182)	(60.3, 161)	(6, 44)	(19, 46)

Normal reference ranges (references from Birmingham Children’s Hospital Clinical Chemistry Laboratory): Calcium 2.2–2.7 mmol/L, phosphate 0.8–1.9 mmol/L, magnesium 0.7–1.0 mmol/L, 25 (OH) vitamin D ≥50 nmol/L; parathyroid hormone (PTH) 15–60 ng/. Legend: CGMP, casein glycomacropeptide; CGMP100, children taking all their protein substitute as casein glycomacropeptide; CGMP50, children taking a combination of casein glycomacropeptide and amino acids; L-AA, amino acids.

**Table 5 nutrients-13-02075-t005:** Median (range) serum bone and urine turnover markers calculated from enrolment, 12, 24, and 36 months for CGMP100, CGMP50 and L-AA groups in girls and boys.

	CGMP100 Boys	CGMP100 Girls	CGMP50 Boys	CGMP50 Girls	L-AA Boys	L-AA Girls
**β-CTX** **μg/L**	1.2 (1.2, 1.6)	1.2 (1, 1.5)	1.2 (1.1, 1.4)	1.2 (1.2, 1.3)	1.4 (1.3, 1.4)	1.2 (0.9, 1.3)
**Bone ALP** **U/L**	86 (76, 95)	103 (92, 106)	125 (114, 131)	108 (76, 116)	85 (75, 95)	83 (46, 97)
**P1NP** **μg/L**	503 (488, 509)	476 (387, 663)	470 (434, 543)	507 (487, 649)	522 (418, 556)	445 (175, 553)
**fDPD** **nmol/L**	178 (68, 307)	114 (71, 338)	207 (91, 227)	147 (98, 265)	157 (96, 247)	107 (93, 114)
**fDPD/Ur Cr** **nmol/mmol**	22 (9, 27)	24 (12, 28)	26 (10, 30)	23 (13, 28)	25 (8, 27)	14 (8, 26)
**fPYD** **nmol/L**	735 (276, 1514)	429 (275, 700)	825 (310, 951)	624 (347, 1134)	615 (331, 876)	413 (290, 436)
**fPYD/Ur Cr** **nmol/mmol**	96 (33, 118)	90 (40, 121)	96 (37, 111)	105 (49, 110)	94 (27, 109)	58 (24, 100)
**Ur Ca/Cr** **mmol/L**	1 (0.4, 1.2)	1.1 (0.8, 1.4)	1.3 (0.7, 1.5)	0.8 (0.4, 1.3)	1.6 (1.3, 2.4)	1.9 (1.3, 2.5)
**Ur Cr** **mmol/L**	12 (1, 15)	6 (5, 11)	8 (7, 9)	8 (8, 10)	10 (8, 16)	7 (6, 8)

Legend: M, males; F, females; CGMP, casein glycomacropeptide; CGMP100, children taking all their protein substitute as casein glycomacropeptide; CGMP50, children taking a combination of casein glycomacropeptide and amino acids; L-AA, amino acids; β-CTX, type 1 collagen β crosslinked C-telopeptide; bone ALP, bone alkaline phosphatase; P1NP, procollagen type 1 N-terminal propeptide; fDPD, urine free deoxypyridinoline; fDPD/Ur Cr, deoxypyridinoline (free)/creatinine ratio; fPYD, urine free pyridinoline; fPYD/Ur Cr, pyridinoline (free)/creatinine ratio; Ur Ca/Cr, urine calcium/creatinine ratio; Ur Cr, urine creatinine. Standard references for children are not available.

**Table 6 nutrients-13-02075-t006:** Median z scores (range) for height, weight, and BMI in the L-AA, CGMP100, and CGMP50 groups, measured annually from enrolment to 36 months in PKU children taking either L-AA, CGMP50, or CGMP100.

**Time** **(Months)**	**L-AA** **Height Z Score** ***n* = 19**	**CGMP50** **Height Z Score** ***n* = 16**	**CGMP100** **Height Z Score** ***n* = 13**
Enrolment (range)	0.2 (−0.2 to 0.8)	−0.1 (−0.6 to 0.6)	−0.1 (−0.4 to 0.3)
36 Months (range)	0.2 (0.0 to 0.5)	0.3 (−0.1 to 0.7)	0.6 (0.1 to 0.7)
	**L-AA** **Weight Z score** ***n* = 19**	**CGMP50** **Weight Z score** ***n* = 16**	**CGMP100** **Weight Z score** ***n* = 13**
Enrolment (range)	0.9 (−1.1 to 3.1)	0.6 (−1.9 to 1.8)	0.4 (−0.6 to 2.3)
36 Months (range)	1.0 (−1.3 to 2.6)	1.2 (−2.4 to2.1)	0.9 (−0.4 to 1.8)
	**L-AA** **BMI Z score** ***n* = 19**	**CGMP50** **BMI Z score** ***n* = 16**	**CGMP100** **BMI Z score** ***n* = 13**
Enrolment (range)	1.2 (−2.5 to 2.0)	0.8 (−0.2 to 2.0)	0.4 (−0.6 to 2.8)
36 Months (range)	1.0 (−0.8 to 2.8)	1.3 (−1.2 to 2.4)	0.9 (−0.9 to 1.8)

## Data Availability

Please contact the corresponding author.

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
