# Peer review of "A Three-Year Longitudinal Study Comparing Bone Mass, Density, and Geometry Measured by DXA, pQCT, and Bone Turnover Markers in Children with PKU Taking L-Amino Acid or Glycomacropeptide Protein Substitutes"

_nutrients, 2021, doi:10.3390/nu13062075_

Round 1

Reviewer 1 Report

Revisions and clarifications made. I believe this publication provides information for clinicians regarding medical food products.

Reviewer 2 Report

The authors have answered the comments and concerns satisfactorily.

This manuscript is a resubmission of an earlier submission. The following is a list of the peer review reports and author responses from that submission.

Round 1

Reviewer 1 Report

Daly et al. present a detailed investigation on the effect of a diet with either amino acids (AA) or casein glycomacropeptide (CGMP-AA) as main protein source – or a combination of both - on the bone density, mass and geometry over 36 months in children with PKU. On the customary Phe-free AA diet, these patients present areal bone mineral density within the normal clinical range but below the population norm. The authors grounded the present study on earlier studies supporting a positive effect of protein consumption on bone and peripubertal bone growth. Studies with PKU animal models have also shown CGMP-AA to increase bone strength. Thus, these previous data and the fact that the use of CGMP compared to AA and their influence on bone mass and density has not been previously studied in PKU children, warranted the present study.

The authors designed and carried out a longitudinal prospective controlled study over 36 months, investigating the efficacy of CGMP–AA compared to AA protein substitute on bone mass, density, geometry and turnover markers in children with PKU. Other data obtained on the same group of children under the same protocol have already been published (Daly et al. 2020). The inclusion criteria for children with PKU, the protocol with separation in 3 groups and the methods appear sound, although the study presents limitations with respect to i) low power due to the small patient numbers in each group, ii) a too short study period, which should ideally be extended to reach the pubertal age for all patients, iii) the inclusion of a healthy control group, and iv) records of exercise. As the results did not support the a priori expectations on a possible advantage of a macropeptide-based protein substitute compared to the AA diet on bone density, mass and geometry, the authors discus these limitations, still concluding that the findings are consistent. Nevertheless, some of the limitations of the study deserve additional attention.

  • Based on the high dependence of BTMs and other markers on age and puberty, among other, the selection of groups with comparable median ages, but different enough to become largely different with respect to % of prepubertal children in the 3 groups, i.e. 38% for CGMP100 and 31% for CGMP50 but 68% in the AA group. This problem could have been solved by extending the study period of > 3 years so that puberty could be reached for a higher proportion of patients in the AA group. As this may not be possible, the authors should improve the comparison of groups. Subgroups of boys and girls are included in some tables/figures, and these subgroups could be considered in each data set. Other patient subgroups of more similar age may be possible. Moreover, the statistical methods used in each table and figure and the p-values obtained should be presented.

  • Results section 3.3: Median weight for height z scores and BMI z scores were above the ideal reference mean indicating an overweight group of children. The authors should better discus the consequences of this finding based on the association between obesity and bone mineral density.

Minor corrections:

  • The authors refer to their previous paper (Daly et al. 2020) for height, weight, body mass index and blood Phe levels in the three groups. Due to the importance of these parameters for the present study, they should be explicitly presented, though properly referenced, also in the present paper.
  • Page 13, line 68: Correct «physioloigical»

Reference:

Daly, A., Evans, S., Pinto, A., Jackson, R., Ashmore, C., Rocha, J. C. and MacDonald, A. (2020) The Impact of the Use of Glycomacropeptide on Satiety and Dietary Intake in Phenylketonuria. Nutrients 12, 2704.

Reviewer 2 Report

Table 3 needs formatting.  Enrolmen on one line t on the line below.  pg. 8 line 261-263, states that there was a significant increase in cortical density using CGMP100.  Doesn't this increase in density decrease the risk of fracture and indicate that the GMP products do indicate an advantage to consuming GMP products??  This was not in the conclusion section.  Also, concerning protein intake pg. 3 line 56-57.  There is a huge range of natural protein intake, 3-30g.  Were there differences in subjects consuming more natural protein??

Author Response

Please fine the manuscript and the supplementary tables
